# 3D Digital Heritage Models as Sustainable Scholarly Resources

**Erik Champion \* and Hafizur Rahaman** 

Discipline of Theatre, Screen and Digital Media, School of Media Creative Arts and Social Inquiry, Faculty of Humanities, Curtin University, GPO Box U1987, Perth, WA 6845, Australia; hafizur.rahaman@curtin.edu.au

\* Correspondence: erik.champion@curtin.edu.au

**Abstract:** If virtual heritage is the application of virtual reality to cultural heritage, then one might assume that virtual heritage (and 3D digital heritage in general) successfully communicates the need to preserve the cultural significance of physical artefacts and intangible heritage. However, digital heritage models are seldom seen outside of conference presentations, one-off museum exhibitions, or digital reconstructions used in films and television programs. To understand why, we surveyed 1483 digital heritage papers published in 14 recent proceedings. Only 264 explicitly mentioned 3D models and related assets; 19 contained links, but none of these links worked. This is clearly not sustainable, neither for scholarly activity nor as a way to engage the public in heritage preservation. To encourage more sustainable research practices, 3D models must be actively promoted as scholarly resources. In this paper, we also recommend ways researchers could better sustain these 3D models and assets both as digital cultural artefacts and as tools to help the public explore the vital but often overlooked relationship between built heritage and the natural world.

**Keywords:** 3D model; virtual heritage; ecosystem; infrastructure

## 1. Introduction

Sustainable digital cultural heritage has been considered a serious national issue in America [1]. Sustaining digital libraries are also a crucial issue [2], and these two concepts share common issues, including a problem with securing long-term funding and ensuring that users continually find the heritage collections (and library collections) useful and worthwhile. Digital reconstructions of cultural heritage have been deployed as showcases for cutting-edge technology and to promote tourism to otherwise remote cultures and distant lands [3,4].

Virtual reality, mixed reality, and augmented reality projects also provide tantalizing new ways of engaging the public with the past [5]. As simulations, scholars might modify them to verify or refute historical hypotheses, testing either data or methods. While the original sites may have existed for hundreds or thousands of years, the digital models that underpin these digital projects have a limited shelf-life, and through designed obsolescence, perceived obsolescence, or the limitations of time, training, and resources, they are seldom successfully deployed in the classroom [6].

As with libraries, museums require both long-term funding and public engagement. However, they have severely limited space and facilities for either exhibition or digitalization, let alone continual funding for new technologies or the time to train staff or teach the public how to best utilize new interaction design technology [7]. The field of digital heritage, with its 3D models and 3D projects, has an added sustainability dilemma: 3D digital heritage models can help promote tourism in remote and endangered areas, therefore helping local businesses, but they can also potentially damage fragile historic places and heritage sites through increased visitation [8].

These are profound meta-issues, but a more immediate yet often overlooked problem is how to help scholars support more appropriate, useful, and required research into both digital heritage technologies and user experience design solutions. For example, 3D models, when used in interactive virtual environments or when integrated into augmented and mixed reality environments, may provide immediate and user-directed simulations communicating how even built heritage sites are predicated on natural features, resources, and ecosystems. Archaeological sites are often prepared to take best advantage of dynamic and seasonal natural resources. Monuments are designed to resist (but ultimately succumb to) natural forces. Sacred buildings often frame constellations and cosmic events. The range and nature of architecture is dependent on local or precious materials. Their remains are palimpsests of human encounters, repeated erosion, personal habits, human-caused pollution, and natural calamities. Game engines and interactive virtual reality technologies can show both these relationships plus changes over time and the effects of human visitation, modulated by the decisions of virtual visitors [9]. A more sustainable development of 3D models to promote the aims of cultural heritage may therefore lead to increased public, institutional, and philanthropic interest, engagement, and investment, in both built heritage and its relationship to the natural environment.

Impetus for more sustainable digital heritage models would ideally be generated by the community of scholars dedicated to the study of digital heritage. After all, education is a major reason for the preservation of digital heritage, according to UNESCO's *Charter on the Preservation of the Digital Heritage* [10]:

> "Preservation of the digital heritage requires sustained efforts on the part of governments, creators, publishers, relevant industries and heritage institutions. In the face of the current digital divide, it is necessary to reinforce international cooperation and solidarity to enable all countries to ensure creation, dissemination, preservation and continued accessibility of their digital heritage . . . The stimulation of education and training programs, resource-sharing arrangements, and dissemination of research results and best practices will democratize access to digital preservation techniques."

In this charter, UNESCO recommends developers, designers, and publishers to work with heritage organizations (such as libraries, museums, and the private sector), professional associations and institutions, and universities (as well as other research organizations) to preserve digital heritage data and to train and share experience and knowledge in a "sustained" fashion. However, there is a critical problem in the scholarship of 3D digital heritage projects [11]. In our initial investigations into this field of scholarship, we did not find many reports building on, corroborating, or verifying previous digital heritage research. In fact, we could not find many digital heritage models directly linked to research projects and openly accessible both as interactive digital experiences and as scholarly resources.

Admittedly, there are successful portals for acquiring free or purchasable 3D heritage models—notable exceptions include Sketchfab, Smithsonian 3D, Europeana, or the Google Arts and Culture- CyArk Open Heritage Project websites. However, there are still far too few instances of scholarly digital heritage projects that are easily accessible to the public or to scholars that are clearly identifiable as scholarly investigations or carefully delineated research projects. There appear to be even fewer scholarly projects that lend themselves to investigation, pedagogical explanation, scholarly verification, design modification, refinement, or amalgamation into larger or newer projects.

As mentioned above, virtual heritage (VH) is commonly used to describe projects that combine virtual reality (VR) and cultural heritage [12,13]. Stone and Ojika [5] defined virtual heritage as "the use of computer-based interactive technologies to record, preserve, or recreate artefacts, sites, and actors of historic, artistic, religious, and cultural significance and to deliver the results openly to a global audience in such a way as to provide formative educational experiences through electronic manipulation of time and space". Various commentators and charters (London, Seville) have also stated that the success of a VH (Virtual Heritage) project depends on 3D models and associated scholarly content [14,15]. Given the above, our starting hypothesis is that there appears to be a dramatic increase in the number of academic papers on 3D digital heritage (especially virtual heritage), but, conversely,

there is a decreasing number of accessible 3D assets [16]. If true, this foretells serious problems in the field of digital heritage as a sustainable scholarly activity, at least if 3D models are considered to be an essential part of scholarly and pedagogical endeavors.

## 2. Method

We ran a literature survey of the 14 proceedings of the last three consecutive publications of major digital heritage events and conferences (Table 1). These were: *The International Society for Virtual Systems and Multimedia (VSMM), Computer Applications and Quantitative Methods in Archaeology (CAA), International Committee of Architectural Photogrammetry (CIPA), The European Mediterranean Conferences (EuroMed)*, and *The Digital Heritage International Congress* (but not *Digital Heritage 2018*). These conferences were chosen as they are arguably major international conferences in digital heritage, and provided online access to the papers. From a total of 1483 conference papers, 264 were selected (Table 2), and 19 were found to contain explicit links to 3D models and related assets (Table 3). The quality of reporting of meta-analysis method (QUORUM) statement presented by Moher et al. [17] was chosen to help select the papers (Figure 1), using the following steps:

*Step 1: Identification and Screening*

1.  Source selection—Popular and renowned international events such as a conference and symposium on digital heritage and allied domains were preferred as initial sources. Proceedings of the last three consecutive publications of these events were selected. This selection, from 2012 to 2017, covered major events, journals, and conferences, such as *The International Society for Virtual Systems and Multimedia (VSMM), Computer Applications and Quantitative Methods in Archaeology (CAA), International Committee of Architectural Photogrammetry (CIPA), The European Mediterranean Conferences (EuroMed)*, and *The Digital Heritage International Congress*.

2.  Retrieval and initial screening—1483 papers from 14 proceedings were collected from their respective digital repositories and publication databases. Articles which contained representative images or references to 3D digital heritage assets were selected for further study. A total of 264 articles were selected at this stage.

*Step 2: Final Screening*

The selected papers were then reviewed in terms of their abstract and a rapid examination of their structure and content in order to:

1.  Exclude irrelevant articles (such as review papers and short survey papers);
2.  Eliminate duplicates (or similar papers published in other proceedings with minimum changes);
3.  Exclude articles on the digitization of paintings and artworks (32 were excluded at this stage).

*Step 3: Review*

1.  Final sorting—The final selection of articles included for the detailed study were: 31 (out of 173) from *VSMM*, 38 (out of 240) from *CAA*, 79 (out of 305) from *CIPA*, 61 (out of 284) from *EuroMed*, and 55 (out of 481) from *Digital Heritage Congress* (a list is attached as a Supplementary Materials).

2.  Study process and analysis—At this phase, the selected articles were studied to find whether they provide any references citing external links or information to:

    (i)    Accessible 3D assets (and the degree of their accessibility);
    (ii)   Accessible video content;
    (iii)  Visual materials (such as VR models, photographs, images of 3D reconstruction, etc.);
    (iv)   Other external resources (if any).

A scheme for recording the study's detailed descriptive data into a database was created in MS Excel. For each article reviewed, required information about the criteria mentioned above was inserted into a spreadsheet and presented in a tabular format.

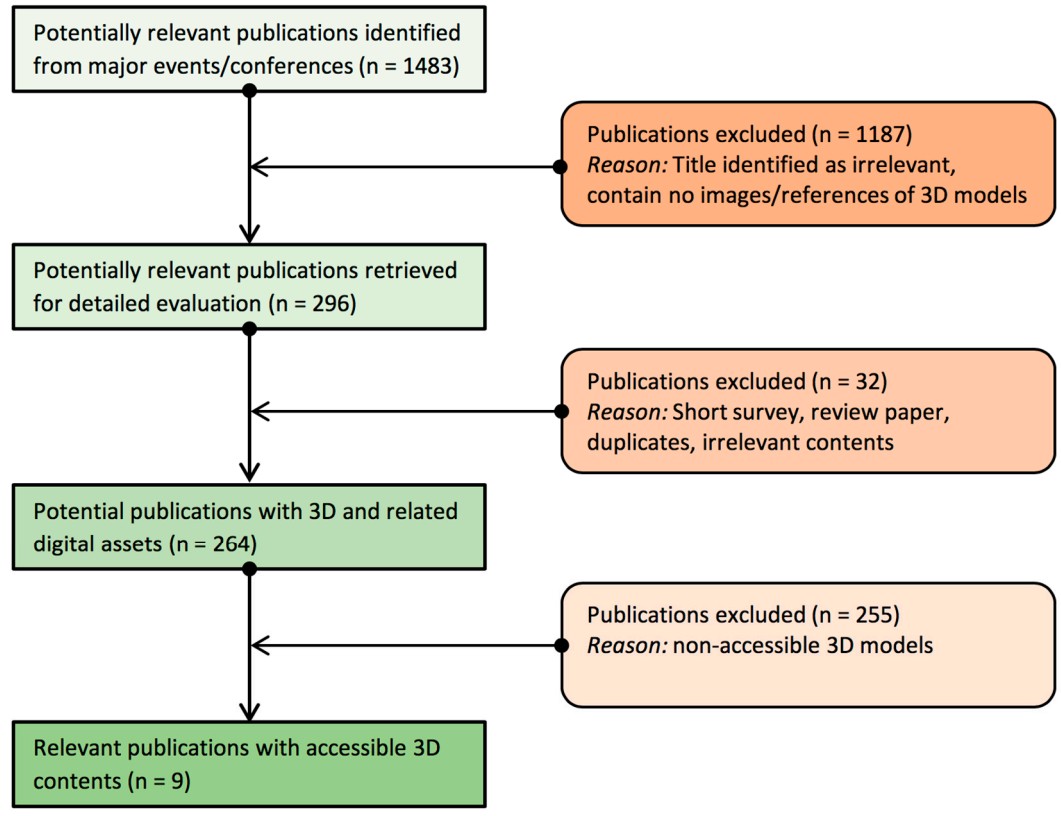

**Figure 1.** QUORUM process.

**Table 1.** 3D heritage conference papers.

| Conference Publications | 2017 | 2016 | 2015 | 2014 | 2013 | 2012 | Total Papers |
|---|---|---|---|---|---|---|---|
| *VSMM* | 55 | 65 | 53 | | | | 173 |
| *CAA* | | | 117 | 73 | 50 | | 240 |
| *CIPA* | 111 | | 82 | | 112 | | 305 |
| *EuroMed* | | 105 | | 84 | | 95 | 284 |
| *Digital Heritage* | | | 270 | | 211 | | 481 |
| TOTAL | 166 | 170 | 522 | 157 | 373 | 95 | 1483 |

**Table 2.** Total articles containing references to 3D models and heritage assets.

| Conference Publications | Total Papers | Mentioning 3D Assets | % |
|---|---|---|---|
| *VSMM 2015–2017* | 173 | 31 | 17.9% |
| *CAA 2013–2015* | 240 | 38 | 15.8% |
| *CIPA 2013, 2015, 2017* | 305 | 79 | 25.9% |
| *EuroMed 2012, 2014, 2016* | 284 | 61 | 21.5% |
| *Digital Heritage 2013, 2015* | 481 | 55 | 11.4% |
| TOTAL | 1483 | 264 | 17.8% |

**Table 3.** Selected papers included in our study.

| Directly Accessible Content | VSMM | CAA | CIPA | EuroMed | Digital Heritage | Total |
|---|---|---|---|---|---|---|
| 3D content | 0 | 1 | 3 | 1 | 4 | **9** |
| Videos | 1 | 2 | 1 | 2 | 6 | **12** |
| Other (VR models, photos, images of 3D models, etc.) | 1 | 4 | 6 | 5 | 17 | **33** |
| 3D assets on accessible websites | 3 | 0 | 5 | 3 | 8 | **19** |

## 3. Results

From a group of 1483 conference papers, we selected 264, accounting for 17.9% of the total papers published in *VSMM, CAA, CIPA, EuroMed*, and *Digital Heritage Congress* from 2012 to 2017. The results of the study, which have been tabulated in Tables 1 and 2, reveal that a significant number of papers (i.e., 17.9%) referred to and contained images of 3D assets or 3D digital models. Contrary to our initial expectations, accessible 3D assets or 3D models were found in only nine papers, i.e., 3.4% of the selected publications.

Of the 264 selected articles, 12 contained external web-links to video content (4.6%) and 33 articles (12.5%) provided external links for other accessible visual material, including VR models, photographs, and images of 3D models. We found 19 articles with external web-links to 3D models. However, not a single one of the links worked at the time of writing this article (last checked: 1 September 2018).

Of the nine articles that provided external links to accessible 3D assets, they all shared four common locations/repositories. These particular nine articles referred to only five external links for storing their 3D assets. The five external links were http://3dicons-project.eu/ (leading to https://www.europeana.eu/), http://dati.comune.bologna.it/3d, www.cyark.org, https://skfb.ly/DtVq, and https://harvest4d.org/?page_id=1367 (last accessed 7 January 2019).

## 4. Discussion

In an upcoming conference paper (to be presented at CAADRIA 2019) we will explore the technical solutions to this problem of the "vanishing virtual"—that is, the dilemma of technology superseding itself [16]. There we will propose a component-based 3D model system that is linked to current infrastructure projects. However, in this article we wish to focus on what we propose is a fairly simple yet barely noticed problem: digital technology has compelled us to seize the historical artefact at one point in time (the time of recording, which is not the time of creation or time of use), and then develop hermetically sealed interfaces and interaction mechanisms around these stillborn 3D copies of the found object or the recorded landscape. This may not initially appear to be a problem—after all, faster processes and bigger, high-resolution screens can exhilarate the senses—but we question whether the accumulative, organic, uniquely situated, and highly dynamic built culture of the past is always best served by apparent precision, speed, and scale.

High-resolution scans and photogrammetric models record one slice of historical action, but they do not necessarily communicate how built culture has responded to natural forces, to human change, or to the pressures of time. Secondly, high-technology demonstrations may impress but do not necessarily engage the public (or even scholars) into exploring process and test theories. A photorealistic 3D digital reproduction (born digital or digital surrogate) is not sufficient by itself [18] for the public to interpret and perceive its cultural significance. Thirdly, high-technology showcases necessitate very expensive equipment, specialized resources, and highly-trained staff (who are often trained in research rather than in public engagement).

Fourthly, such advanced equipment can exact a high price not only from the public or private purse, but also from natural resources (in terms of both energy and materials). Merely being in the cloud also has an energy cost: for example, in 2015 Google consumed roughly the same energy as

the city of San Francisco [19] and in the same year the Internet was predicted to contribute roughly the same amount to global emissions as air travel [20]. A 2018 *Nature* article warned that by 2030, thanks to an explosion in data centers and increasing shared social media content, Information and Communication Technology could consume up to 21% of total global energy [21]. Nor is moving digital heritage content to increasingly powerful smartphones an ideal situation. By 2020, 5 billion phones could be in circulation, and the rare earth metals they use could run out in 20 to 50 years. Meanwhile, the extraction of iron, aluminum, and copper, not to mention gold and tin, have already resulted in catastrophic spills, deforestation, and toxic poisoning [22].

Digital heritage has a price. While digital heritage projects are likely to contribute only a small percentage to this energy consumption and to the depletion of rare metals and minerals, we suggest that the environmental resources consumed should be explicitly considered in the design of any major project. We also suggest that digital heritage as an educational medium and as a channel for communication and collaboration among scholars across the world, with access to wildly varying resources, be considered.

Moreover, we suggest that digital heritage needs to understand the contribution of 3D models to the field not only as finished products but also as pedagogical and theoretical building blocks. While the design and deployment of high-technology showcases has its place, there should also be room for the design, sharing, and redesign of simpler objects, scripts, and related digital heritage media that can be modified and improved on by not just a single team but also by a community. This pathway may prove to be more sustainable for the digital models themselves, as well as more beneficial to the aims and objectives of the research community and more effective in disseminating and promoting cultural heritage awareness and understanding. In other words, the academic community should put more emphasis on sharing, critiquing, reusing, and improving the elements of virtual heritage projects, rather than relying on overall projects inside proprietary, locked frameworks.

Simple mechanisms to aid the wider sharing of models, infrastructures, scripts, and media might be to design competitions, grants, and prizes to award to digital heritage projects and communities based on their sharing, verification, modification, and improvement of others' original models and data. Secondly, contributions to open access infrastructures, repositories, and tools should be recognized and supported by universities and related research organizations, while tools, projects, and papers that advance these goals could also be specifically recognized. This includes new forms of publications that emphasize collaboration and feedback around 3D models as specific scholarly resources and as components of scholarly arguments. As far as we know, none of the surveyed digital heritage conferences specify awards or recognition for papers and projects that share 3D models as scholarly assets and scholarly arguments, or for projects involving not only the design but also the evaluation and preservation of digital models. We believe this is not only feasible but also likely to increase the direct linking of publicly accessible models.

In terms of scholarly understanding, there is surprisingly little written and debated about the 3D digital heritage model considered as a learning tool or experimental device rather than as a finished (if virtual) object. Simply put, 3D models are not yet fully integrated into scholarly discourse [23]. At an instrumental level, more uptake is required to establish file formats [11,24] that 'travel' and to develop more tools and frameworks (such as http://www.meshlab.net/) in order to allow content to move between different programs. This would help the modification and collaboration of models. Increasing the use of metadata and Linked Open Data tools [25] and frameworks would help increase the visibility and probably also the usage of digital heritage models. However, the single most effective way to increase public access to 3D digital models, we argue, is to develop various levels of copyright specifically for 3D content that allow owners to share various levels of resolution (or precision) of their 3D models and 3D data [26], along with incentives for them to share various levels of resolution and precision of those models.

## 5. Conclusions

This survey not examined how digital heritage conference papers have addressed the issue of sustainability per se, but it also indicated that the 3D models associated with these papers are not typically seen as worthy of preservation in their own right, which leads us to question both the sustainability of digital heritage as a serious scholarly activity (how can the discipline evolve if we cannot verify each other's data?) and the pedagogical value of these 3D models. However, the problems are so widespread that it appears to be foremost a problem of infrastructure, or more accurately, a problem raised by not having suitable infrastructure. There have been impressive European Union (EU) infrastructure-related projects (ARIADNE, CARARE, 3D–ICONS, Scottish Ten, etc.) and relevant National Endowment of the Humanities projects in the United States, but accessible and well-maintained links to the related 3D assets need to be integrated into academic publication and dissemination systems.

We propose, following the London Charter and others, that 3D models must be recognized as a scholarly resource [27,28]; however, we suggest that there is a key element missing from such charters: a framework or set of guidelines to help create and maintain a robust infrastructure that underpins 3D digital heritage models. There could also be tools and procedures run by digital heritage conferences to provide a framework to view 3D models in relation to the articles (some journals have already begun to explore this, though conferences, to our knowledge, have not).

A further aspect of this article was to suggest that the relationship of built heritage to natural ecosystems has not been fully addressed by digital heritage models. Only when we tackle the challenge of communicating the dynamic and environmentally situated nature of built cultural heritage will we be able to communicate not only the visual effects but also the principles of both the scholarly research underpinning the digital heritage simulation and the sustainability issues of the heritage site itself. Just as the scholarly publication system needs to see itself as more of an evolving scholarly digital ecosystem, which can be continually tested, debated, and updated, so too should the digital heritage project be considered to be not merely a standalone object or a finished product, but a component of process. For how can digital heritage fulfil the noble aims of cultural heritage if it cannot even maintain, preserve, and sustain itself?

**Supplementary Materials:** The following are available online at http://www.mdpi.com/2071-1050/11/8/2425/s1.

**Author Contributions:** Conceptualization of the overall project was devised and managed by E.C., but the surveying and analysis of papers and related 3D assets was conducted by H.R. E.C. wrote the initial draft and overall paper, based on the survey study by H.R. who also provided the initial workflow illustration and table, feedback and corrections on the paper draft and also some of the observations and recommendations.

**Funding:** This research received no external funding.

**Acknowledgments:** We would like to thank Curtin University for support and internal funding.

**Conflicts of Interest:** The authors declare no conflict of interest.

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
