# Peer review of "3D Digital Heritage Models as Sustainable Scholarly Resources"

_sustainability, doi:10.3390/su11082425_

Round 1
Reviewer 1 Report
This study deals with sustainable digital heritage, in particular on the poor link between scholarly activity and accessible 3D assets. The topic is interesting.
The paper is well written, clear and mainly exhaustive from the methodological point of view.
However, in my opinion it could be reviewed taking into consideration some European and Italian projects that represent a valid case of application of 3D models and dissemination of digital heritage. As European example, the paper refers only to the Europeana Project (because it is mentioned in one of the 1483 papers analyzed), but there are other projects financed by European Union or national funds in which there are valid examples of virtual heritage, i.e. PARTHENOS, ARIADNE, V-Must, SICaR, IPERION CH, E-RIHS, etc). Evidently, these cases have never been presented in the five conferences cited by authors in the examined period, perhaps published in peer reviewed journals. Could be correct to specify this aspect.
Author Response
This study deals with sustainable digital heritage, in particular on the poor link between scholarly activity and accessible 3D assets. The topic is interesting.
The paper is well written, clear and mainly exhaustive from the methodological point of view.
Response 1: Thank you.
However, in my opinion it could be reviewed taking into consideration some European and Italian projects that represent a valid case of application of 3D models and dissemination of digital heritage. As European example, the paper refers only to the Europeana Project (because it is mentioned in one of the 1483 papers analyzed), but there are other projects financed by European Union or national funds in which there are valid examples of virtual heritage, i.e. PARTHENOS, ARIADNE, V-Must, SICaR, IPERION CH, E-RIHS, etc). Evidently, these cases have never been presented in the five conferences cited by authors in the examined period, perhaps published in peer reviewed journals. Could be correct to specify this aspect.
Response 2: Thank you w are aware of these projects and have been involved with some and mentioned in other publications. But this is to do with acces to the models per se, not the EU projects themselves. We will include in the conclusion remarks on these projects and link to relevant papers and projects where possible, inclkuding the new ARIADNE PLUS, which sounds very promising. We will also add one or two projects from the US that we know of.
As we only have 10 days to revise for 3 reviewer comments, our changes will be limited.
Reviewer 2 Report
The paper present an overview of digital 3D contents in Cultural Heritage. Although the topic is important and hot, the paper does not present a real conclusion and does not go in deepen the problem. The solution to the problem is not easy but the authors don't even try to highlight different solutions.
I suggest to enlarge the discussion, introducing some solutions or thoughts about possible variants.
Author Response
The paper present an overview of digital 3D contents in Cultural Heritage. Although the topic is important and hot, the paper does not present a real conclusion and does not go in deepen the problem. The solution to the problem is not easy but the authors don't even try to highlight different solutions.
I suggest to enlarge the discussion, introducing some solutions or thoughts about possible variants.
Response 1: We will include in the conclusion remarks on these projects and link to relevant papers and projects where possible, including the new ARIADNE PLUS, which sounds very promising. We will also add one or two projects from the US that we know of and link to recent publications where we published recommendations
As we only have 10 days to revise for 3 reviewer comments, our changes will be limited to highlighting key recommendations.
(x)Moderate English changes required
Response 2: We will proof-read the English and change sentences that may not be as clear as they could be. We will also rewrite the introduction and conclusion.
Regards
Is the research design appropriate?
Are the methods adequately described?
We will also update the method section and improve the clarity of the table. Will our above changes address these two criteria, as it is not clear to us how they link to the text-based comments?
Reviewer 3 Report
The article presents an interesting topic related to the sustainability and possibilities of reusing 3D digital models. The authors draw attention to the limited accessibility of 3D models scanned in many projects and campaigns for third parties. The article touches on a weighty subject and is certainly interesting for the journal.
In the case of this article, I have only one but very important consideration. I am a practitioner of 3D scanning and I have been struggling with a similar problem many times - how to put 3D objects in the internet alone and not ready presentations in VR. It is worth noting that the independent 3D digital object has several stages of its life:
- the model created directly from the measurement data - it is characterized by the highest possible fidelity of the mapping (it has the largest documentary value) but also usually does not represent the original object in full due to the shortcomings of the available measurement methods (some holes in geometry representation, not equally illuminated texture, etc.),
- simplified model for visualization via the Internet - significantly simplified and partially modeled manually (low documentation value),
- model prepared for specific purposes / presentations - different levels of simplification with the subjective layer (average documentation value).
What exactly should be shared? How to indicate what is given from the measurement (documentation) and what about the creation (in the best case hypothesis). How do you do this by sharing only the model and descriptive metadata?
I would like to extend the content of the article with its consideration.
Author Response
The article presents an interesting topic related to the sustainability and possibilities of reusing 3D digital models. The authors draw attention to the limited accessibility of 3D models scanned in many projects and campaigns for third parties. The article touches on a weighty subject and is certainly interesting for the journal.
Response 1: Thank you.
In the case of this article, I have only one but very important consideration. I am a practitioner of 3D scanning and I have been struggling with a similar problem many times - how to put 3D objects in the internet alone and not ready presentations in VR. It is worth noting that the independent 3D digital object has several stages of its life:
- the model created directly from the measurement data - it is characterized by the highest possible fidelity of the mapping (it has the largest documentary value) but also usually does not represent the original object in full due to the shortcomings of the available measurement methods (some holes in geometry representation, not equally illuminated texture, etc.),
- simplified model for visualization via the Internet - significantly simplified and partially modeled manually (low documentation value),
- model prepared for specific purposes / presentations - different levels of simplification with the subjective layer (average documentation value).
What exactly should be shared? How to indicate what is given from the measurement (documentation) and what about the creation (in the best case hypothesis). How do you do this by sharing only the model and descriptive metadata?
Response 2: We only had 10 days notice to revise the paper, and will not be able to address this fully in the current paper as it is focussed on conference publication. However we share your concern and wish to work on a project allowing for different levels of resolution and file size, controlled by the content owner and linked to levels of access so that the content owner can determine what levels of quality can be linked to groups, and do this we do need to tackle questions of precision and accuracy. We wonder if work like the PhD thesis of Valeria Vitale on levels of accuracy in interpretation can be matched to levels of accuracy and whether institutions can suggest criteria for people to create metadata levels to, and whether Linked Open Data can provide links to related work and a way to measure currency, interpretation support (in terms of number of people or quality of related institutes) and whether Prof Bernard Frischer in Indiana or the group at UCLA have developed such a measure (as their CVRLab did say there was an advisory board). The question is whether they had heuristics for this and whether CIPA ICOMOS or CAA or another organization could put this into practice with publications or dynamic web links. Also, could Meshlab be modified to create level of interpretation levels aand link via a shared ontology to related models, and to their perceived accuracy, and degree of completeness?
Sadly, we don't think we have the time or space to address this issue fully in this paper.
Round 2
Reviewer 2 Report
I appreciate the effort of the authors in only 10 days
Author Response
thank you.
Reviewer 3 Report
The authors did not make any significant amendments to the article. In its current form, it is a report of the current state. It does not carry any scientific novelty.
Author Response
>The authors did not make any significant amendments to the article. In its current form, it is a report of the current state. It does not carry any scientific novelty.
Hello
this survey has never been done before.
we do provide recommendations and link to an upcoming paper with more recommendations.
we have a set word limit and 10 days to provide reply 1 and 5 days to reply 2.
we did adjust the paper but the specific recommendations were out of scope.